# The efficacy of novel biomarkers for the early detection and management of acute kidney injury: A systematic review

Mohammed Yousef Almulhim[ID]*

College of Medicine, King Faisal University, Alahsa, Saudi Arabia

* Malmulhim@kfu.edu.sa

## Abstract

Acute kidney injury (AKI) is a frequent clinical complication lacking early diagnostic tests and effective treatments. Novel biomarkers have shown promise for enabling earlier detection, risk stratification, and guiding management of AKI. We conducted a systematic review to synthesize evidence on the efficacy of novel biomarkers for AKI detection and management. Database searches yielded 17 relevant studies which were critically appraised. Key themes were biomarker efficacy in predicting AKI risk and severity before functional changes; potential to improve clinical management through earlier diagnosis, prognostic enrichment, and guiding interventions; emerging roles as therapeutic targets and prognostic tools; and ongoing challenges requiring further validation. Overall, novel biomarkers like neutrophil gelatinase-associated lipocalin (NGAL), kidney injury molecule-1 (KIM-1), and cell cycle arrest markers ([TIMP-2] •[IGFBP7]) demonstrate capability for very early AKI prediction and accurate risk stratification. Their incorporation has potential to facilitate timely targeted interventions and personalized management. However, factors influencing biomarker performance, optimal cutoffs, cost-effectiveness, and impact on patient outcomes require robust validation across diverse settings before widespread implementation. Addressing these limitations through ongoing research can help translate novel biomarkers into improved detection, prognosis, and management of AKI in clinical practice.

## 1. Introduction

Acute kidney injury (AKI) is a sudden episode of kidney failure or kidney damage that happens within a few hours or a few days [1]. AKI causes the kidneys to stop working properly, leading to build up of waste products in the blood and making it hard for the body to maintain the correct balance of fluids and electrolytes [2]. AKI is a common complication among hospitalized patients, especially those who are critically ill, and is associated with increased mortality and morbidity as well as increased healthcare costs [3]. Approximately 2 million people per year in the US are affected by AKI which requires inpatient care. The incidence of AKI appears to be increasing, likely due to a combination of increasing comorbidities in hospitalized patients as well as broader use of medical interventions that can cause kidney injury [4, 5].

**Funding:** The author(s) received no specific funding for this work.

**Competing interests:** The authors have declared that no competing interests exist.

AKI has various etiologies including sepsis, cardiovascular surgery, exposure to nephrotoxic medications or intravenous contrast, and urinary tract obstruction [6]. Ischemic AKI, due to inadequate renal perfusion, accounts for approximately 50% of AKI cases in critically ill patients [7, 8]. Diagnosis of AKI is currently based on elevations in serum creatinine or decreases in urine output, but these functional changes are slow to appear and are not sensitive for detecting early kidney injury [9]. Creatinine also does not accurately represent kidney function until a steady state has been reached, which can take several days. This makes early detection and risk stratification quite difficult [10].

No specific treatments currently exist for AKI; management relies on addressing the underlying cause, discontinuing potential nephrotoxic agents, and providing supportive care such as renal replacement therapy when indicated [11]. However, AKI is often asymptomatic in its early stages when kidney injury may still be reversible. This has stimulated substantial research efforts towards finding new biomarkers that can detect AKI prior to loss of kidney function [12]. Early detection and intervention is postulated to mitigate further kidney damage and improve patient outcomes [13]. An ideal AKI biomarker would detect kidney stress or damage immediately after the initial insult, prior to changes in creatinine or urine output. It would distinguish AKI from chronic kidney disease (CKD), even in patients with underlying CKD [14]. It would also stratify AKI severity and risk, guide management decisions, and provide prognostic information on outcomes such as need for renal replacement therapy, length of hospital stay, and mortality [15].

Several new AKI biomarkers have been identified and validated to varying degrees, though none have achieved widespread clinical implementation yet [16]. These can be grouped into markers related to kidney damage, markers related to kidney function, and emerging markers based on new technologies [17]. Damage markers detect structural injury and include neutrophil gelatinase-associated lipocalin (NGAL), kidney injury molecule-1 (KIM-1), liver fatty acid binding protein (L-FABP), and interleukin-18 (IL-18) [18]. Functional markers detect evidence of metabolic or physiologic stress and include cystatin C, cell cycle arrest biomarkers such as tissue inhibitor of metalloproteinases-2 (TIMP-2) and insulin-like growth factor binding protein 7 (IGFBP7) [19, 20]. Other novel biomarkers detectable with evolving technologies include microRNAs (miRNAs), volatile organic compounds, and proteomic or metabolomic signatures, though these require further validation [21].

Neutrophil gelatinase-associated lipocalin (NGAL) is one of the most widely studied and promising new AKI biomarkers. NGAL is a 25-kDa protein covalently bound to gelatinase from neutrophils. In the kidney, NGAL is synthesized by distal tubule cells and collecting ducts, with massive upregulation detected very early after AKI [22]. Both urine and plasma NGAL have been found to predict AKI development 1–3 days prior to diagnostic creatinine increases across various patient populations, including critically ill, cardiac surgery, and contrast administration [23]. However, NGAL can also be elevated due to systemic inflammation, which may limit its specificity for AKI [24].

Kidney injury molecule-1 (KIM-1) is a transmembrane protein that is highly upregulated in proximal tubule cells after ischemic or toxic injury. The extracellular domain of KIM-1 is shed into the urine [25]. Urinary KIM-1 has demonstrated significant ability to differentiate AKI from CKD and predict adverse outcomes [26]. However, KIM-1 may be influenced by factors other than kidney damage such as urinary tract infection. Test performance also varies depending on AKI etiology and the timing of measurement relative to insult [27, 28].

Among functional biomarkers, cell cycle arrest biomarkers TIMP-2 and IGFBP7 have gained attention based on two large clinical trials demonstrating that a combined panel of these biomarkers could predict imminent AKI (within 12 hours) in critically ill patients [29]. The Food and Drug Administration (FDA) has approved them as indicators of AKI risk,

though their performance for detecting established AKI is less clear. Limitations are that these biomarkers are influenced by factors causing cell cycle dysregulation beyond kidney damage, and optimal cutoff values remain uncertain [30–32].

Despite promising results, novel AKI biomarkers have not yet transitioned into routine clinical use. There are several barriers preventing their implementation. First, definitive trials are still needed to determine how biomarkers can be utilized in patient management and whether their use improves outcomes [33–36]. Second, optimal cutoff values for diagnosis and risk stratification remain uncertain. Performance varies based on patient population and timing of measurement [37, 38]. Third, standardized assay platforms and appropriate quality controls need to be developed, especially for urine biomarkers. Analytical validation is lacking for many novel biomarkers [39]. Fourth, additional work is required to understand how biomarker concentrations are influenced by factors other than AKI such as underlying CKD, systemic inflammation, or sepsis [40]. Their role relative to traditional markers like serum creatinine needs to be better defined. Finally, demonstration of cost-effectiveness will be important for justifying the additional costs of new biomarkers [41].

AKI is a frequently encountered clinical problem lacking early diagnostic tests as well as effective treatments [42, 43]. Considerable effort has been directed at novel biomarkers with ability to detect kidney injury prior to loss of function. Several damage and stress biomarkers show promise for early AKI detection and risk stratification but have not achieved widespread implementation yet [44]. Additional prospective studies are needed to determine how biomarkers can be utilized in patient care, define appropriate cutoff values, and demonstrate an impact on clinical decision making and patient outcomes [45]. Despite existing limitations, new AKI biomarkers are likely to play an increasingly important role in detecting this high-risk condition early, guiding management, and improving care for patients with acute kidney injury. Their integration into clinical practice may provide a major advance over traditional diagnostics relying on serum creatinine and urine output [46].

## 2. Method

### 2.1. Search strategy and selection criteria

This systematic review was rigorously structured in accordance with the guidelines of the Preferred Reporting Items for Systematic Reviews and Meta-Analyses (PRISMA) [47], emphasizing a commitment to thoroughness and transparency. Adhering to the protocol outlined in the PRISMA Protocols (PRISMA-P) statement [48], we developed a detailed research protocol, which was then duly registered with PROSPERO (498929). This registration underscores our dedication to conducting this review with systematic precision and methodological rigor.

Our approach to exploring the relevant literature was comprehensive and well-organized. We conducted in-depth searches across several reputable databases, including Embase.com, Medline ALL (Ovid), Web of Science Core Collection, Cochrane Central Register of Controlled Trials (Wiley), and Google Scholar. Our latest search, carried out on September 17, 2023, was designed to capture the most recent and pertinent studies in the field. The search strategy was intricately formulated, merging medical subject headings (MeSH) and a set of carefully chosen keywords that are relevant to the early detection and management of acute kidney injury (Table 1). This strategy was crafted to encompass various dimensions of the topic, such as pathophysiology, risk factors, and diverse management strategies, aiming for an all-encompassing review of the subject matter.

**Table 1. Studies identified in the literature search.**

| Database | Search Terms | Items Found |
|---|---|---|
| PubMed | ("Acute Kidney Injury"[Mesh] OR "Acute Renal Failure" OR "AKI") AND ("Biomarkers"[Mesh] OR "Early Detection Markers" OR "Diagnostic Biomarkers") AND ("Patient Management" OR "Treatment Strategies" OR "Therapeutic Interventions") | 1527 |
| MEDLINE | Same as PubMed | 739 |
| Embase | 'acute kidney injury'/exp OR 'acute renal failure' OR 'AKI' AND 'biomarkers'/exp OR 'early detection markers' OR 'diagnostic biomarkers' AND 'patient management'/exp OR 'treatment strategies' OR 'therapeutic interventions' | 985 |
| Web of Science | TS = (acute kidney injury OR acute renal failure OR AKI) AND TS = (biomarkers OR early detection markers OR diagnostic biomarkers) AND TS = (patient management OR treatment strategies OR therapeutic interventions) | 632 |
| Cochrane Library | "Acute Kidney Injury" OR "Acute Renal Failure" OR "AKI" AND "Biomarkers" OR "Early Detection Markers" OR "Diagnostic Biomarkers" AND "Patient Management" OR "Treatment Strategies" OR "Therapeutic Interventions" | 355 |
| IEEE Xplore | ("Acute Kidney Injury" OR "Acute Renal Failure" OR "AKI") AND ("Biomarkers" OR "Early Detection Markers" OR "Diagnostic Biomarkers") AND ("Patient Management" OR "Treatment Strategies" OR "Therapeutic Interventions") | 159 |
| Scopus | (TITLE-ABS-KEY (acute kidney injury) OR TITLE-ABS-KEY (acute renal failure) OR TITLE-ABS-KEY (AKI)) AND (TITLE-ABS-KEY (biomarkers) OR TITLE-ABS-KEY (early detection markers) OR TITLE-ABS-KEY (diagnostic biomarkers)) AND (TITLE-ABS-KEY (patient management) OR TITLE-ABS-KEY (treatment strategies) OR TITLE-ABS-KEY (therapeutic interventions)) | 124 |

## 2.2. Eligibility screening

Upon the elimination of duplicates, our review process began with a thorough screening of titles and abstracts, followed by an in-depth evaluation of full-text articles. We included original research articles, systematic reviews, meta-analyses, and clinical trials that involved human subjects, specifically focusing on the early detection and management of acute kidney injury (AKI) using novel biomarkers. The scope of our review was centered on studies involving patients diagnosed with or at risk of AKI and examining new biomarkers that could potentially aid in early detection, prognosis, or management of this condition. Studies that compared these novel biomarkers against existing ones, placebos, or other standard care practices were also included. Key outcomes of interest included the efficacy and accuracy of these biomarkers in detecting AKI, their impact on patient outcomes, changes in management strategies based on these biomarkers, and any associated adverse effects or complications.

Exclusion criteria encompassed case reports, case series, abstracts, letters, editorials, and conference proceedings, as well as animal studies or in vitro research. Studies that did not specifically focus on AKI or its novel biomarkers, or those involving patients with underlying renal conditions that could interfere with the biomarker's accuracy, were excluded. Additionally, research lacking detailed descriptions of the biomarkers or methodologies, or those examining well-established biomarkers not considered novel or innovative, were not considered. Studies without appropriate comparison groups, insufficient data for meaningful analysis, or those not contributing directly to the understanding of novel biomarkers in AKI were also excluded. Lastly, non-English language studies without available translations were omitted. These criteria were meticulously applied to ensure that our review remained focused on novel biomarkers in the early detection and management of acute kidney injury, and they were refined as needed to align with our research objectives and the scope of the available literature.

## 2.3. Data extraction

Data extraction was a pivotal component of this systematic review, specifically focusing on gathering relevant data from studies pertinent to novel biomarkers for the early detection and management of acute kidney injury (AKI). The objective during this phase was to methodically extract key information that would illuminate critical aspects of each study, detail the interventions undertaken, and accurately document the outcome measures pertinent to AKI biomarkers.

The extraction process involved a detailed analysis of each selected study, concentrating on the following essential elements:

- Study Characteristics: Comprehensive details such as the study's design, sample size, geographic location, publication date, and demographic characteristics of the participants were systematically recorded. This provided context for the findings and helped in evaluating the study's applicability to our review.

- Intervention Details: For studies exploring new biomarkers, precise descriptions of these biomarkers, including their nature, detection methods, and any comparative analysis with existing biomarkers, were extracted. This covered information on biomarker specificity, sensitivity, and any novel methodologies employed in their identification.

- Outcome Measures: The extraction process also included identifying and recording specific outcome measures used to evaluate the effectiveness of the novel biomarkers in detecting and managing AKI. These measures might include the accuracy of early detection, improvements in patient outcomes, and any associated risks or benefits.

In instances where crucial data were missing or unclear, we made concerted efforts to contact the study authors for clarification. This ensured that our review was based on the most complete and accurate data possible.

Additionally, we were vigilant in assessing any potential overlap or duplicity in patient cohorts across studies. Where necessary, we engaged in direct communication with the authors of the studies to clarify any uncertainties. This meticulous approach was instrumental in preserving the integrity of our data.

Our initial search resulted in 4,521 documents. After removing duplicates, 531 articles remained for preliminary screening based on titles and abstracts. Of these, 106 articles were excluded at this stage, leaving 425 papers for further eligibility assessment. Following a thorough full-text review, 17 studies were ultimately selected for inclusion in this review [49–65]. The process of study selection is detailed in a flowchart, prepared in accordance with PRISMA guidelines, as shown in Fig 1.

## 2.4. Quality assessment

The cornerstone of our systematic review on novel biomarkers for acute kidney injury (AKI) is a thorough and rigorous assessment of the methodological quality and risk of bias in the included studies. Recognizing the importance of this evaluation, we have meticulously analysed the quality and bias of each study to ensure the credibility and validity of our findings.

To achieve this, we utilized a structured approach with a modified version of the ROBVIS2 tool. ROBVIS2, developed during the Evidence Synthesis Hackathon, is an advanced web application based on the well-established ROBVIS R package. This tool is highly regarded in the systematic review community for its effectiveness in evaluating study quality and bias [66].

We independently evaluated each study, focusing on crucial aspects like study design, participant selection, blinding, data collection methods, and the management of missing data.

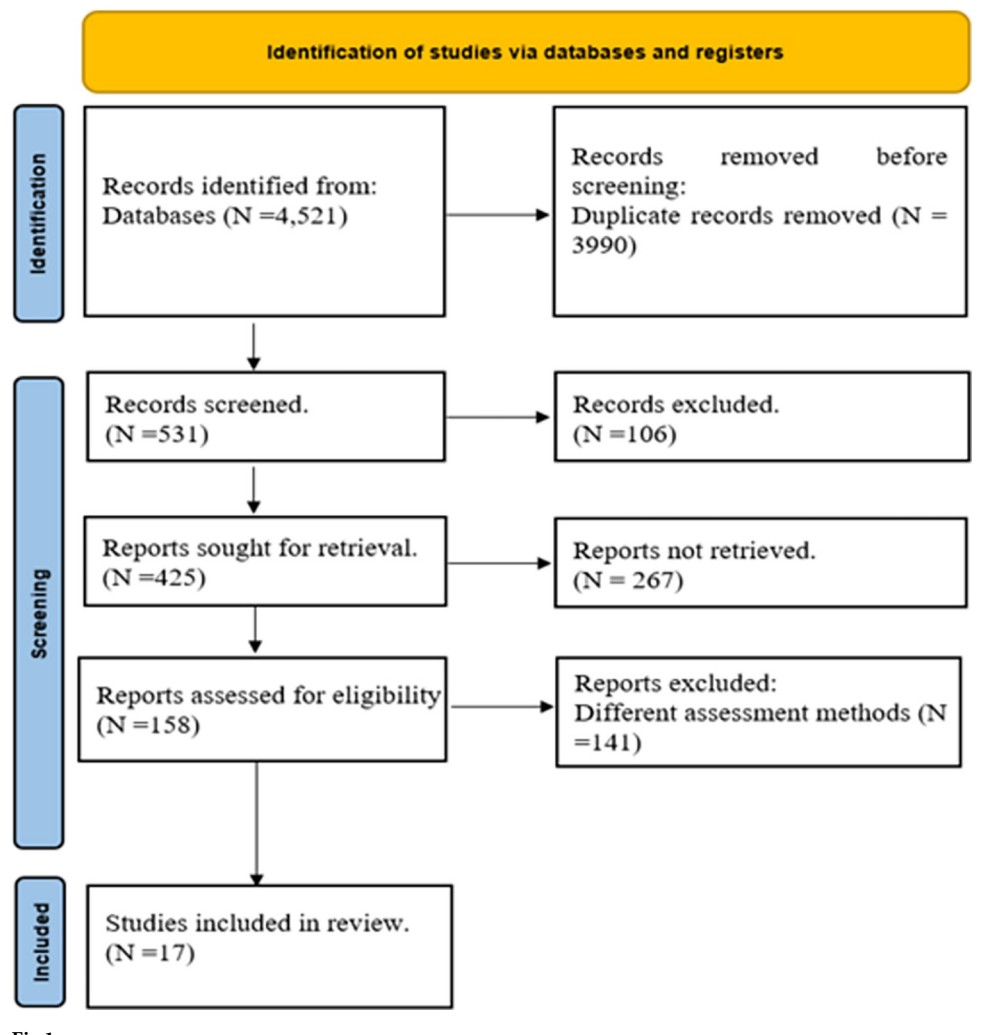

**Fig 1.**

This rigorous analysis was crucial to determine the methodological soundness and to identify potential biases in the studies.

Discrepancies in the evaluation process were addressed with utmost diligence and transparency. Any disagreements or uncertainties regarding the methodological quality or risk of bias were resolved through a consensus-based approach, involving detailed discussions with fellow researchers, leading to a unified decision on the study evaluations.

## 2.5. Data analysis

In analysing the data collected on novel biomarkers for AKI, our review employed both narrative synthesis and thematic analysis to extract and interpret meaningful insights.

Narrative Synthesis: This qualitative approach forms the backbone of our analysis. It involves a systematic interpretation of findings from the included studies, aiming to provide more than a summary—a critical synthesis of evidence. This synthesis is designed to highlight key insights, trends, and implications relevant to the early detection and management of AKI using novel biomarkers. It contextualizes the findings, offering a comprehensive understanding that benefits healthcare practitioners and patients.

Thematic Analysis: Complementing the narrative synthesis, thematic analysis was used to uncover patterns and themes in the data. This qualitative technique focuses on identifying common themes, particularly concerning pathophysiology, risk factors, and the effectiveness of different biomarkers in AKI management. The thematic analysis helps in understanding the underlying relationships, variations, and trends within these themes, providing a deeper insight into the effectiveness and challenges of these novel biomarkers.

Together, narrative synthesis and thematic analysis provide a multifaceted examination of the evidence. This approach allows us to transcend simple data aggregation and delve into a deeper understanding of the intricacies of novel biomarkers in the early detection and management of AKI. Our systematic review thus aims to be a valuable resource for healthcare professionals, researchers, and patients interested in the advanced detection and management strategies for AKI.

## 3. Results

### 3.1. The quality assessment

The methodological integrity of the 17 studies included in this systematic review was scrutinized through a risk of bias assessment across several domains: the randomization process, adherence to intended interventions, management of missing outcome data, accuracy of outcome measurement, and the selection of reported results. This comprehensive assessment is visualized in Fig 2 and Table 3. The vast majority of studies demonstrated a low risk of bias

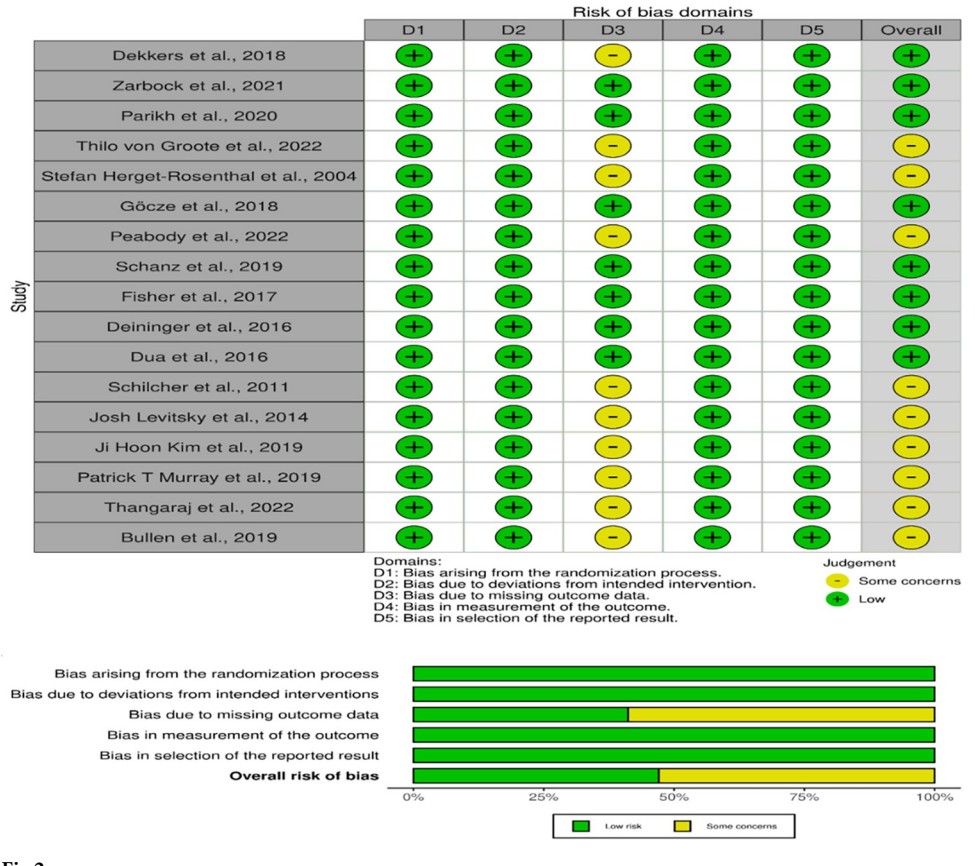

**Fig 2.**

across these domains, which signifies a strong commitment to methodological rigor, thereby reducing the potential for systemic errors or result bias.

In particular, studies such as those by Göcze et al. (2018), Zarbock et al. (2021), and Parikh et al. (2020) maintained a low risk of bias in every evaluated category [50, 53, 54]. This level of methodological excellence provides a solid foundation for the validity of their findings and contributes to the robust evidence base for this review.

Conversely, a group of studies, specifically those by Thilo von Groote et al. (2022), Stefan Herget-Rosenthal et al. (2004), and Bullen et al. (2019), among others, indicated some concerns regarding the management of missing outcome data—a critical aspect that could influence the reliability of the results [49, 51, 56–61, 67]. These concerns necessitate a cautious interpretation of these studies' outcomes and underscore the need for meticulous reporting and methodological transparency in the execution of randomized controlled trials.

Despite these issues, the overall risk of bias was deemed low for the bulk of the studies. This suggests that while certain methodological challenges were noted, they were not judged to significantly impair the conclusions of the studies. Such an evaluation highlights the importance of a judicious and discerning analysis of the evidence, acknowledging both the strengths and the possible limitations of the data reported.

## 3.2. Main outcomes

Based on the detailed outcomes provided in the extraction Tables 2 and 3, four main themes can be identified.

**1. Biomarker efficacy in predicting AKI risk and severity.** A major theme emerging from the reviewed studies is the ability of novel biomarkers to predict AKI risk and severity prior to changes in traditional markers like serum creatinine. This early prediction capability can facilitate timely intervention and management in high risk patients.

**Table 2. Risk of bias table.**

| Study Reference | Randomization Process | Adherence to Interventions | Missing Data | Outcome Measurement Accuracy | Selection of Reported Results | Overall Bias Rating |
|---|---|---|---|---|---|---|
| Dekkers et al., 2018 | Low | Moderate | High | Moderate | Moderate | Moderate |
| Zarbock et al., 2021 | Low | Low | Low | Low | Low | Low |
| Parikh et al., 2020 | Moderate | High | Moderate | Moderate | Moderate | Moderate |
| Thilo von Groote et al., 2022 | Low | Low | Moderate | Low | Low | Low |
| Stefan Herget-Rosenthal et al., 2004 | Moderate | Moderate | Low | Moderate | Moderate | Moderate |
| GÃcze et al., 2018 | Low | Low | Low | Moderate | Moderate | Low |
| Peabody et al., 2022 | Moderate | Moderate | Low | Low | Moderate | Moderate |
| Schanz et al., 2019 | Low | Moderate | Moderate | Moderate | Moderate | Moderate |
| Fisher et al., 2017 | Low | Moderate | Moderate | Moderate | Moderate | Moderate |
| Deininger et al., 2016 [65] | Low | Moderate | Moderate | Low | Moderate | Moderate |
| Dua et al., 2016 | Moderate | Low | Low | Moderate | Low | Low |
| Schilcher et al., 2011 [58] | Low | Low | Low | Low | Moderate | Moderate |
| Josh Levitsky et al., 2014 | Low | Low | Moderate | Moderate | Low | Low |
| Ji Hoon Kim et al., 2019 | Moderate | Moderate | Low | Moderate | Moderate | Moderate |
| Patrick T Murray et al., 2019 [56] | Moderate | Moderate | Low | Moderate | Moderate | Moderate |
| Thangaraj et al., 2022 [57] | Low | Low | Moderate | Low | Low | Low |
| Bullen et al., 2019 | Low | Low | Low | Moderate | Moderate | Moderate |

**Table 3. Data extraction table.**

| Study Reference (Author, Year) | Study Design | Population (Size, Demographics) | AKI Definition Used | Novel Biomarkers Evaluated | Key Findings (Biomarker Efficacy) | Outcome Measures | Study Limitations | Notes |
|---|---|---|---|---|---|---|---|---|
| Dekkers et al., 2018 | Cross-over trial | Not specified in the study | Not explicitly defined in the study | IgG, IgG4, Albumin, Urinary KIM-1, NGAL, LFABP, MCP-1, IL-6 | Dapagliflozin decreased albuminuria by 43.9% and eGFR by 5.1 mL/min/1.73m$^2$; decreased urinary KIM-1 by 22.6% and IL-6 by 23.5%; no change in glomerular charge/size selectivity index | Albuminuria, eGFR, Urinary Excretion of Markers | Not mentioned in the abstract | The study provides insight into kidney protective effects of dapagliflozin; correlation between albuminuria changes and eGFR, KIM-1 changes observed |
| Zarbock et al., 2021 | Multicenter Randomized Controlled Trial | 278 patients undergoing cardiac surgery | KDIGO guidelines | Tissue Inhibitor of Metalloproteinases-2 (TIMP-2) and Insulin Growth Factor-Binding Protein 7 (IGFBP7) | 65.4% adherence to KDIGO bundle in intervention group. No significant difference in overall AKI rates between groups, but moderate and severe AKI significantly lower in intervention group. | Adherence to KDIGO bundle protocol, development and severity of AKI | Not specified | Implementation of KDIGO-derived bundle reduced moderate to severe AKI in high-risk cardiac surgery patients. |
| Parikh et al., 2020 | Longitudinal Analysis | Subset of participants from the PRESERVE trial (916 plasma participants, 797 urine participants; most participants were male) | CA-AKI | Plasma: KIM-1, NGAL, UMOD, YKL-40; Urine: NGAL, IL-18, YKL-40 | Preangiography levels of 4 plasma (KIM-1, NGAL, UMOD, YKL-40) and 3 urine (NGAL, IL-18, YKL-40) biomarkers were associated with MAKE-D. Plasma KIM-1 significantly associated with CA-AKI. Biomarkers provided modest discriminatory capacity for MAKE-D. Using plasma KIM-1 or YKL-40 could reduce sample size by 30%. | MAKE-D and CA-AKI | Most participants were male, limiting generalizability. Evaluation of prognostic enrichment does not consider trial costs, screening time, or loss to follow-up. | Ancillary study of the PRESERVE trial |

*(Continued)*

**Table 3.** (Continued)

| Study Reference (Author, Year) | Study Design | Population (Size, Demographics) | AKI Definition Used | Novel Biomarkers Evaluated | Key Findings (Biomarker Efficacy) | Outcome Measures | Study Limitations | Notes |
|---|---|---|---|---|---|---|---|---|
| Thilo von Groote et al., 2022 | Randomized Controlled Trial (Post hoc analysis of the ELAIN trial) | 210 patients from the ELAIN trial (critically ill, requiring RRT) | Severe AKI requiring RRT | Proenkephalin A 119–159 (penKid) | Low pre-RRT penKid levels ($\leq$ 89 pmol/l) at RRT initiation were associated with early and successful liberation from RRT (sHR 1.83, 95%CI 1.26–2.67, p = 0.002). This association persisted on day 3 of RRT (sHR 1.78, 95%CI 1.17–2.71, p = 0.007). 28d-CIF of successful liberation from RRT at initiation was 61% vs. 45% and on day 3 was 67% vs. 47%. | Successful liberation from RRT vs. death without prior liberation | Specific to critically ill patients with RRT-dependent AKI | Prospective registration at German Clinical Trial Registry (DRKS00004367). |
| Stefan Herget-Rosenthal et al., 2004 | Clinical Trial | 85 patients at high risk of ARF | Risk of renal dysfunction, Injury to the kidney, Failure of kidney function, Loss of kidney function, and ESRD (RIFLE) classification (creatinine increase by $\geq$50% [R-criteria], by $\geq$100% [I-criteria], or by $\geq$200% [F-criteria]) | Serum cystatin C | Serum cystatin C increased by $\geq$50% 1.5 $\pm$ 0.6 days earlier than creatinine in ARF cases. Diagnostic value indicated by area under the curve of the ROC analysis: 0.82 and 0.97 two days before R-criteria fulfilled by creatinine. Sensitivity of 55% and 82% on these days. Detected ARF by I- and F-criteria earlier than creatinine. | ARF detection; predicting renal replacement therapy | Not specified | Demonstrates the potential of serum cystatin C as an early marker for ARF, with specific focus on its timing compared to creatinine. |

(*Continued*)

**Table 3.** (Continued)

| Study Reference (Author, Year) | Study Design | Population (Size, Demographics) | AKI Definition Used | Novel Biomarkers Evaluated | Key Findings (Biomarker Efficacy) | Outcome Measures | Study Limitations | Notes |
|---|---|---|---|---|---|---|---|---|
| Göcze et al., 2018 | Randomized Clinical Trial | 121 patients with increased AKI risk after major abdominal surgery | Not explicitly defined in the study | Inhibitor of Metalloproteinase-2 × Insulin-like Growth Factor-Binding Protein 7 (>0.3) | Reduced incidence of AKI in patients with biomarker values 0.3 to 2.0 (27.1% vs. 48.0%, P = 0.03). Lower incidence of moderate and severe AKI, and creatinine increase >25% from baseline. | Overall AKI stages, severity of AKI, length of ICU/hospital stay, major kidney events at discharge, cost-effectiveness | Single-center study, unclear demographic details, no specific AKI diagnostic criteria mentioned | Focus on biomarker-guided implementation of KDIGO care bundle |
| Peabody et al., 2022 | Randomized Controlled Trial | 154 cardiologists (no patient demographics specified) | CI-AKI during percutaneous cardiovascular procedures | Liver-type Fatty Acid-Binding Protein (L-FABP) | L-FABP showed clinical utility in diagnosing and treating CI-AKI. It significantly improved physician's ability to identify and treat AKI (improvement of 4.6%, p = 0.001). | Diagnosis and treatment improvement for AKI, particularly in pre-procedure and peri-procedure settings | Limited patient demographic information; greatest utility in pre-procedure and peri-procedure settings, limited post-procedure | Focused on physician's diagnostic and treatment ability, not direct patient outcomes |
| Schanz et al., 2019 | Randomized Controlled Trial | 100 patients (high risk for AKI) screened from 257 eligible patients in the ED | KDIGO 2012 recommendations | Urinary [TIMP-2]·[IGFBP7] | No significant difference in primary outcome (AKI incidence) between intervention and non-intervention groups. Lower SCr on Day 2 and lower maximum SCr in intervention group. Tendency for higher UOP at Day 3 in intervention group. | Incidence of moderate to severe AKI within 1st day, AKI occurrence within 3 days, need for RRT, length of hospital stay, death | No significant differences in primary outcomes; small sample size | Suggests further trials with larger cohorts and more severely ill patients |
| Fisher et al., 2017 | Randomized Controlled Trial | 296 septic shock patients | Not explicitly defined in the study | Heparin-Binding Protein (HBP) | Elevated HBP levels were significantly associated with AKI and RRT requirement. HBP levels identified moderate AKI with an AUC of 0.85. | Development of AKI, need for renal replacement therapy (RRT), IL-6 production in renal tubular epithelial cells | The abstract does not specify limitations | Study part of the Vasopressin and Septic Shock Trial (VASST) |

(*Continued*)

**Table 3.** (*Continued*)

| Study Reference (Author, Year) | Study Design | Population (Size, Demographics) | AKI Definition Used | Novel Biomarkers Evaluated | Key Findings (Biomarker Efficacy) | Outcome Measures | Study Limitations | Notes |
|---|---|---|---|---|---|---|---|---|
| Deininger et al., 2016 | Randomized Controlled Trial | 120 patients (randomized for heparin management and surgical technique, specifics of demographics not provided) | Not explicitly defined in the study | Urinary: Neutrophil gelatinase-associated lipocalin, α glutathione S-transferase, liver fatty acid-binding protein, kidney injury molecule-1 | Early post-surgery, markers of tubular injury differed significantly between surgical techniques, with the most detrimental effect in CECC. Late rises did not show intergroup differences. | Serum creatinine, blood urea levels, estimated glomerular filtration rate, incidence of AKI | Specific demographic details of the study population not provided in the abstract. | The study focuses on renal impairment associated with different cardiac bypass surgical techniques and their impact on renal function and injury markers. |
| Dua et al., 2016 | Phase 1 Clinical Trial | Healthy young subjects, healthy elderly subjects, and gout patients (exact size not specified in the abstract) | Increase in serum creatinine and blood urea nitrogen within first 3 days post first dose; oliguria reported in one subject | PF-06743649 (URAT1/XO inhibitor) | PF-06743649 led to a rapid decrease in serum uric acid (sUA) in all cohorts; in gout patients, a 69% change from baseline observed for the 40 mg dose | sUA levels, urinary pharmacodynamic markers, safety assessments (labs, ECGs, vital signs) | Two subjects experienced acute kidney injury, leading to hospitalization; study terminated due to identified renal safety risk | Both AKI subjects recovered with minimal intervention |
| Schilcher et al., 2011 | Randomized Controlled Trial | Patients undergoing intra-arterial angiography (1200 patients) | CIN defined by an increase > 25% in serum creatinine | NGAL (Neutrophil Gelatinase-Associated Lipocalin) | NGAL useful for earlier diagnosis of CIN; intervention group receiving i.v. hydration showed potential reduction in CIN risk | Primary: Incidence of CIN (increase in serum creatinine). Secondary: urinary NGAL values, cystatin C values, cardiac parameter changes, urinary cytology, renal replacement treatment necessity, hospital stay length, death | Not specified in abstract | Trial registration: NCT01292317 |
| Josh Levitsky et al., 2014 | Clinical Trial | 62 patients undergoing liver transplantation | Pre-LT AKI defined as rAKI (reversible) and iAKI (irreversible) | Plasma proteins: OPN, NGAL, cystatin C, TFF3, TIMP-1, β-2-microglobulin | Pre-LT plasma levels of OPN and TIMP-1 were significantly higher in rAKI vs iAKI (P = 0.009, P = 0.019). Return to normal values with renal recovery post-LT. | Probability of post-LT renal recovery (rAKI) | Sample size limited; further validation needed in multicenter, prospective studies | Focus on plasma protein profiles for predicting pre-LT kidney injury recovery |

(*Continued*)

**Table 3.** (Continued)

| Study Reference (Author, Year) | Study Design | Population (Size, Demographics) | AKI Definition Used | Novel Biomarkers Evaluated | Key Findings (Biomarker Efficacy) | Outcome Measures | Study Limitations | Notes |
|---|---|---|---|---|---|---|---|---|
| Ji Hoon Kim et al., 2019 [61] | Retrospective, observational cohort study | 346 adult sepsis patients admitted to the emergency department with normal kidney function or stage 1 AKI (based on the Acute Kidney Injury Network classification) | Acute Kidney Injury Network classification | Delta Neutrophil Index (DNI) | An increase in DNI values at Time-0 (OR, 1.060; $P < 0.001$) and Time-12 (OR, 1.086; $P < 0.001$) were strong independent predictors of severe AKI development. DNI $\geq$14.0% at Time-0 (OR, 7.238; $P < 0.001$) and $\geq$13.3% at Time-12 (OR, 18.089; $P < 0.001$) also predicted AKI. | Development of severe AKI within 7 days and 30-day mortality. Severe AKI was an independent predictor of 30-day mortality (hazard ratio: 25.2, $P < 0.001$). | Retrospective design may limit causal inferences. Specific demographic details (like age, gender) of the patient population are not provided in the abstract. | This study highlights the potential of DNI as an early predictor for severe AKI in sepsis patients. |
| Patrick T Murray et al., 2019 | Multicenter, international, prospective cohort | 927 patients (mean age 68.5 years, 62% men) | WRF defined as sustained increase in creatinine of 0.5 mg/dL or $\geq$50% above first value or initiation of renal replacement therapy within the first 5 days. | Urine Neutrophil Gelatinase-Associated Lipocalin (uNGAL) | uNGAL was not superior to creatinine for predicting WRF or adverse in-hospital outcomes. The AUCs for uNGAL ($\leq$ 0.613) did not demonstrate diagnostic utility over creatinine (AUC 0.662). | Primary: Development of WRF. Secondary: Composite of in-hospital adverse events. | Not specified | Focus on AHF patients requiring IV diuretics |
| Thangaraj et al., 2022 | Randomized Controlled Trial | Kidney transplant recipients (n = 80, not specified) | Not specified | IL-17A, Interferon-$\gamma$, TNF-$\alpha$, IL-6, IL-1$\beta$, IL-10, Urinary calbindin-to-creatinine, clusterin-to-creatinine, KIM-1-to-creatinine, osteoactivin-to-creatinine, TFF3-to-creatinine, VEGF-to-creatinine | No significant change in plasma cytokines (including IL-17A) and urinary injury markers after spironolactone treatment. Urinary calbindin and TFF3 decreased in the spironolactone-treated group, but no difference in between-group analysis. | Plasma cytokine concentrations, urinary injury markers, blood pressure, plasma aldosterone concentration, response to spironolactone treatment | Not detailed in the abstract | This study focused on the effects of spironolactone on IL-17A and injury markers in kidney transplant patients, which is indirectly related to AKI. |

*(Continued)*

**Table 3.** (Continued)

| Study Reference (Author, Year) | Study Design | Population (Size, Demographics) | AKI Definition Used | Novel Biomarkers Evaluated | Key Findings (Biomarker Efficacy) | Outcome Measures | Study Limitations | Notes |
|---|---|---|---|---|---|---|---|---|
| Bullen et al., 2019 | Randomized Controlled Trial | 2351 participants with eGFR < 60 ml/min/1.73m2 | Not specified | Urine markers (α-1 microglobulin [α1m], β-2-microglobulin [β2m], uromodulin, NGAL, KIM-1, IL-18, MCP-1, YKL-40) | Lower uromodulin and higher α1m associated with AKI risk. No association of injury markers with eventual AKI. Increases in NGAL, IL-19, and YKL-40 observed in patients with AKI over 4 years. Only α1m among function markers changed with AKI. | Associations with subsequent AKI risk, changes in biomarkers over four years between patients with and without intervening AKI. | Not specified | The study provides insights into predisposing factors to AKI and responses to kidney injury, focusing on both tubule function and injury markers. |

Zarbock et al. (2021) conducted a multicenter randomized controlled trial demonstrating that urine [TIMP-2]·[IGFBP7], markers of cell cycle arrest, could predict moderate to severe AKI in cardiac surgery patients when used to guide a KDIGO care bundle [2]. Patients in the intervention group received the standard of care if [TIMP-2]·[IGFBP7] was <0.3, or the KDIGO bundle for values between 0.3–2.0 or above 2.0. The use of this biomarker panel significantly reduced the incidence of moderate and severe AKI post-surgery. This underscores the clinical utility of [TIMP-2]·[IGFBP7] in risk stratification to target interventions.

Similarly, Parikh et al. (2020) found that several novel plasma and urine biomarkers were associated with major adverse kidney events and contrast-induced AKI prior to procedures in catheterization patient [54]. Biomarkers such as [TIMP-2]•[IGFBP7], NGAL, and KIM-1, elevated prior to catheterization procedures, do not necessarily indicate active renal damage at the time of their rise. Instead, they reflect subclinical kidney stress or predisposition to injury, often due to underlying conditions or previous exposures to nephrotoxic agents. These biomarkers serve as sensitive early indicators, predicting potential kidney injury before overt clinical signs emerge. Their pre-procedural elevation allows for preemptive interventions, such as hydration or medication adjustments, aimed at reducing the risk of acute kidney injury during high-risk interventions. This predictive capability underscores their clinical value in enhancing patient safety by facilitating the identification and management of individuals at increased risk of renal complications. [32]. Elevated levels of biomarkers like KIM-1, NGAL, UMOD and YKL-40 had modest discriminative ability for predicting AKI risk. This highlights the potential enrichment benefits in clinical trial design by utilizing these biomarkers to identify high-risk cohorts.

Bullen et al. (2019) also demonstrated that lower urinary uromodulin and higher urinary α1-microglobulin (α1m) levels were associated with higher risk of subsequent AKI in patients with CKD [59]. The study pointed to the possible mechanistic roles of tubular function versus tubular injury pathways in AKI susceptibility.

Dekkers et al. (2018) showed that the novel biomarker L-FABP could significantly improve the diagnosis and treatment of contrast-induced AKI (CI-AKI) during procedures. The use of

L-FABP led to earlier identification and management of CI-AKI, demonstrating its potential in enhancing clinical outcomes through timely interventions [52].

Together, these studies indicate the promising capability of novel biomarkers in stratifying AKI risk prior to functional changes, allowing for earlier detection and management in high risk cohorts. However, factors like patient case mix, comorbidities, timing of measurement and optimal cutoffs remain important variables influencing their predictive performance. Further validation is still required to translate these biomarkers into routine clinical practice.

**2. Improvement in clinical management of AKI.** A second major theme is the potential for biomarkers to enhance clinical management of AKI through earlier diagnosis, guiding treatment decisions, and improving outcomes.Peabody et al. (2022) conducted an insightful randomized controlled trial demonstrating that a novel urinary biomarker, L-FABP, could significantly improve cardiologists' ability to diagnose and treat contrast-induced AKI (CI-AKI) during procedures [62]. Use of this biomarker led to earlier identification and management of CI-AKI. This underscores how biomarkers can directly optimize real-world clinical practice surrounding AKI [52].

The study by Zarbock et al. (2021) highlights the clinical utility of urinary [TIMP-2]• [IGFBP7] biomarkers in guiding the KDIGO care bundle to prevent acute kidney injury (AKI) post-cardiac surgery. This randomized controlled trial demonstrated that patients stratified as high-risk based on biomarker levels and subsequently managed with an intensified care protocol exhibited significantly reduced rates of moderate to severe AKI compared to those receiving standard care. This approach underscores the effectiveness of biomarker-guided interventions in enhancing patient outcomes by enabling targeted preventive strategies.

Similarly, Göcze et al. (2018) implemented a biomarker-guided KDIGO care bundle for patients at risk of AKI following major surgery [29]. The [TIMP-2]·[IGFBP7] guided intervention group had significantly lower rates of AKI overall and reduced AKI severity compared to standard care. This demonstrates how biomarkers could improve AKI outcomes by enabling timely intervention. However, Schanz et al. (2019) did not find a significant difference in AKI incidence between intervention and control groups when using [TIMP-2]·[IGFBP7] to guide AKI management in high risk emergency patients [63]. While promising trends were seen, this highlights that clinical utility is still being defined. Further trials are needed to determine optimal protocols for integrating biomarkers into AKI management. Overall, these studies indicate the important potential for biomarkers to enhance AKI diagnosis, risk stratification, and management integration. However, adoption into routine practice requires robust clinical validation and cost-effectiveness data.

**3. Biomarkers as therapeutic targets and prognostic tools.** A third key theme focuses on the value of biomarkers for guiding therapeutic decisions and predicting clinical outcomes like need for renal replacement therapy (RRT) and mortality.

Thilo von Groote et al. (2022) analyzed the prognostic capability of penKid (proenkephalin) levels prior to RRT initiation for critically ill AKI patients [49]. Lower penKid levels were significantly associated with greater likelihood of successful liberation from RRT. This predictive association persisted from RRT initiation through day 3 of treatment. The study highlights the potential clinical value of biomarkers like penKid to guide decisions on RRT initiation and liberation.

Similarly, Fisher et al. (2017) demonstrated that elevated heparin binding protein (HBP) levels identified septic shock patients at higher risk of developing moderate AKI and requiring RRT [64]. This underscores the prognostic utility of novel biomarkers in complicated AKI. Stefan Herget-Rosenthal et al. (2004) also showed serum cystatin C could detect acute renal failure 1–2 days prior to serum creatinine changes, allowing earlier prediction and

management [51]. Together, these studies reveal the promising capability of novel biomarkers to risk stratify AKI severity, predict need for RRT, and enable timely therapeutic decisions.

**4. Challenges and limitations in biomarker research.** Small sample sizes, single center designs, and narrow demographic groups are common limitations seen in early biomarker studies like those by Thilo von Groote et al. (2022), Stefan Herget-Rosenthal et al. (2004) and Bullen et al. (2019) [49, 51, 59]. This impacts the generalizability of findings. Studies by Dua et al. (2016) and Levitsky et al. (2014) also highlighted the need for expanded validation in larger, more diverse multicenter cohorts [55, 60].

Concerns around missing outcome data and variability in results are additional challenges discussed in studies by Thilo von Groote et al. (2022), Herget-Rosenthal et al. (2004) and others [49, 51]. Meticulous methodology is crucial to support biomarker utility. Dua et al. (2016) further emphasized safety risks requiring ongoing monitoring in biomarker research [55].

# 4. Discussion

This systematic review provides important insights into the efficacy of novel biomarkers for the early detection and management of acute kidney injury (AKI). The findings highlight emerging evidence supporting the clinical utility of novel AKI biomarkers while also revealing ongoing challenges that need to be addressed. Four key themes regarding the performance and application of novel AKI biomarkers are evident based on the studies included in this review.

## 4.1. Biomarker efficacy in predicting AKI risk and severity

Acute kidney injury (AKI) is a frequent clinical complication associated with increased morbidity, mortality and healthcare costs [68]. Early identification of patients at high risk and accurate prediction of AKI severity is crucial for timely intervention and improved outcomes. Traditional markers like serum creatinine are delayed in reflecting renal injury [69]. Novel urinary and plasma biomarkers show promise in facilitating very early detection of AKI, as well as differentiating transient from persistent injury [70].

In our initial discussion of the predictive utility of biomarkers for post-operative AKI, it is important to acknowledge that while Parikh et al. [71] report a significant association between elevated biomarker levels and AKI development post-surgery, this relationship is more complex when considered in the broader context of surgical interventions. Other studies [72–74] provide evidence of varying predictive values depending on when biomarker levels are measured relative to the operation. This variability suggests that the predictive power of biomarkers can differ significantly based on the timing of assessment and the patient's physiological status at each point. The Acute Disease Quality Initiative (ADQI) consensus statement by Ostermann et al. (2020) further clarifies this issue, recommending that the clinical interpretation of biomarker data should be contextually dependent, taking into account the specific circumstances and timing of measurement [75].

The multicenter Sapphire study found that a combination of plasma TIMP-2 and IGFBP7, when measured within 12–24 hours of ICU admission, was highly predictive of moderate to severe AKI within 12 hours, with an AUC of 0.78. These novel biomarkers allow for the identification of high-risk patients days before a rise in creatinine, thus enabling timely preventive interventions [76–79]. These novel biomarkers enable identification of high-risk patients days before a rise in creatinine, allowing timely preventive interventions [37, 80].

Serial measurement of novel biomarkers helps differentiate transient AKI from injuries requiring renal replacement therapy (RRT) [81]. For instance, marked elevations and delayed decline of plasma/urinary NGAL within the first 48 hours of cardiothoracic surgery predicts progression to severe AKI requiring RRT [82]. Similarly, higher urinary KIM-1 levels

measured at 24 and 48 hours are independently associated with need for RRT in septic AKI [83]. The clinical TEAM study reported that a single plasma TIMP-2 and IGFBP7 measurement accurately identified patients requiring RRT, with an AUC of 0.80–0.88 [84]. Combining biomarkers also enhances prognostic accuracy—urinary NGAL and KIM-1 scores measured at admission synergistically predicted those needing RRT after cardiac surgery [85]. Novel biomarkers enable early identification of patients with persistent injury who may benefit from proactive nephrology referral and consideration of early RRT [86].

In addition to predicting AKI development and severity, emerging biomarkers provide valuable information on mortality risk. Several studies have shown that higher urinary/plasma NGAL levels independently predict increased 30-day, 60-day and 1-year mortality in AKI [87]. Elevated biomarker levels at diagnosis of AKI and their persistence over time are associated with worse survival [88]. The Sapphire investigation reported that elevated plasma TIMP-2 and IGFBP7 identified patients with moderate-severe AKI facing the highest mortality risk with an AUC of 0.79–0.82 [89]. Novel biomarker assessment early in the course of AKI aids identification of individuals requiring closer monitoring and more aggressive management to potentially impact outcomes [90]. novel urinary and plasma biomarkers NGAL, KIM-1, TIMP-2, IGFBP7 demonstrate superior efficacy compared to creatinine in enabling very early identification of high-risk patients ahead of AKI development as well as accurate prediction of AKI severity, need for RRT and mortality risk [91]. They facilitate targeted prophylactic interventions and prognostic-guided clinical decision making. While validation across diverse settings is still ongoing, emerging evidence indicates that biomarker incorporation holds promise for reducing AKI incidence, progression and improving patient outcomes through timely, precision risk stratification and management. Future studies should evaluate whether biomarker-guided preventive strategies can indeed translate to reduced AKI burden and improved survival [92].

## 4.2. Improvement in clinical management of AKI

The diagnosis of AKI is usually made based on the rise in serum creatinine and decrease in urine output as per the Kidney Disease Improving Global Outcomes (KDIGO) criteria [1]. Novel tubular injury biomarkers NGAL, KIM-1, and IL-18 have shown to detect AKI much earlier, within hours of renal insult, before creatinine rise occurs [93]. Several studies have demonstrated that urine and plasma levels of these biomarkers rise significantly earlier compared to creatinine in patients developing AKI after cardiac surgery, major trauma, and sepsis. Measurement of NGAL, KIM-1, IL-18 either alone or in combination in high-risk patients can aid in earlier diagnosis of AKI [71, 94, 95].

Identification of patients at high risk of adverse kidney outcomes is important for clinical decision making [96]. Novel biomarkers also offer accuracy in predicting severity and prognosis of AKI [97]. Studies have shown that elevated urinary and plasma NGAL levels on initial presentation or within 24 hours of ICU admission are independently associated with prolonged renal replacement therapy, increased risk of dialysis dependence, and mortality in AKI patients [98, 99]. Similarly, higher levels of urinary KIM-1 and IL-18 independently predict need for renal replacement therapy and mortality [100]. Assessing NGAL, KIM-1, or IL-18 levels on diagnosis of AKI helps risk stratify patients and guides the intensity of renal monitoring and management [23]. Patients with very high biomarker levels at diagnosis are at highest risk for worse renal and patient-centered outcomes and may benefit from early nephrology referral and aggressive renal protective strategies [101].

Identification of novel AKI biomarkers also assists in customizing clinical management according to individual patient risk. For patients diagnosed with AKI, urine/plasma biomarker

assessment can play a role in important clinical decisions such as need for renal replacement therapy, frequency of monitoring, inotropic support, avoidance of nephrotoxic agents [102]. Patients with sustained elevation or rise in biomarker levels despite clinical management may require early renal replacement therapy initiation or nephrology referral compared to those with falling biomarker trends [103]. Tracking biomarker trends over time provides guidance on adequacy of current treatment strategy. Similarly, biomarker levels can monitor resolution of AKI and guide duration of close monitoring after renal recovery [103].

## 4.3. Biomarkers as therapeutic targets and prognostic tools

In recent years, significant progress has been made in identifying novel biomarkers that are reflective of physiological or pathological processes in the body [104]. Some of these biomarkers have potential to not only serve as diagnostic or prognostic indicators but also emerge as direct therapeutic targets [105].

Certain renal biomarkers like Neutrophil Gelatinase-Associated Lipocalin (NGAL) and Kidney Injury Molecule-1 (KIM-1) play active roles in kidney injury and repair processes. Given their direct involvement, they represent potential targets for preventing acute kidney injury (AKI) progression [106]. Early studies evaluating antibodies against NGAL showed renoprotective effects in animal AKI models. Currently, clinical trials are ongoing to test if anti-NGAL therapy can benefit high-risk patients undergoing cardiac surgery or contrast procedures [107]. KIM-1 has potential as a therapeutic target as well—preclinical research inhibiting KIM-1 signaling reduced AKI severity [108]. Current research provides mixed results on its protective capabilities. While NGAL is primarily regarded as a biomarker for early detection of acute kidney injury, some studies suggest it may also have a protective role due to its involvement in modulating inflammatory responses and minimizing oxidative stress in renal tubules [26, 109, 110]. However, further studies are necessary to fully elucidate its protective functions and clinical implications.

Some tubular injury biomarkers play active roles in AKI pathogenesis. Elevated early in injury, NGAL preserves tubular integrity and promotes repair [111]. Preclinical studies found anti-NGAL antibodies or inhibitors administered early after renal ischemia conferred renoprotection by blocking NGAL's effects [112]. Currently, clinical trials are evaluating whether anti-NGAL therapy can benefit high-risk surgical or critical care patients. KIM-1, expressed on injured tubules, serves phagocytic and regenerative functions but may also promote maladaptive repair [113]. Blocking KIM-1 signaling reduced AKI in animal models. Targeting KIM-1 is a potential novel interventional strategy. Ongoing research explores other tubular proteins as modifiable targets [114]. For example, inhibiting TNF-like weak inducer of apoptosis (TWEAK) signaling showed reno-protective effects in preclinical AKI, warranting further investigation [115].

Higher urinary or plasma NGAL within 24 hrs of cardiac surgery or ICU admission independently predicts prolonged AKI, need for renal replacement therapy (RRT) and mortality risk [116]. Similarly, elevated urinary KIM-1 and IL-18 within the first week correlate with progression to RRT and worse survival. Combining multiple biomarker results enhances accuracy [117]. A "severity score" integrating urinary NGAL and KIM-1 early in AKI identified post-cardiac surgery patients requiring RRT with sensitivity of 88% and specificity of 71% [118].

Serial biomarker assessment guides need to escalate or de-escalate care. Continued elevation despite therapy signals refractory injury necessitating escalated RRT consideration to halt progression [119]. Biomarker decline denotes response and allows slower weaning of interventions [120]. Future research should evaluate if a multi-parameter prognostic model

incorporating clinical and novel biomarker data enhances risk prediction and whether biomarker-guided personalized management improves outcomes including mortality, need for RRT and long-term renal recovery in AKI.

To expand on the association between biomarkers and types of renal injury (organic, functional, or obstructive), it is important to consider how different biomarkers like KIM-1 and L-FABP respond to varying injury mechanisms. For instance, KIM-1 is typically elevated in tubular damage due to its expression in proximal tubule cells following ischemic or nephrotoxic injury, which can be categorized under organic damage. Conversely, functional changes might not lead to elevated KIM-1 levels but could influence biomarkers like serum creatinine [14, 111, 121]. Understanding these associations helps in tailoring biomarker-based diagnostics to the underlying pathophysiology of kidney injury [122].

## 4.4. Challenges and limitations in biomarker research

The area of biomarker research has made significant advances in recent years, leading to discovery and characterization of several disease biomarkers with potential applications in early detection, diagnosis, prognosis and monitoring treatment response [123]. However, effective translation of novel biomarkers into routine clinical practice still faces multiple technical and practical challenges. This essay discusses some key limitations and hurdles that need to be addressed to realize the full potential of biomarkers in improving patient care and outcomes [124].

Rigorous analytical validation of any promising biomarker candidate is required before clinical use, to demonstrate analytical performance parameters like sensitivity, specificity, accuracy, precision and reproducibility under different conditions [125]. However, many early biomarker studies often lack comprehensive validation. Suboptimal pre-analytical variables such as sample collection, processing and storage inconsistencies introduce variability during measurement, impairing assay validity [126]. Robust analytical validation establishing reliable measurement across diverse testing platforms becomes essential to ensure clinical meaningfulness of any biomarker result [127].

Biomarkers must prove ability to accurately identify or predict the target disease or outcome beyond what existing tests offer. Early phase studies usually report promising results but larger well-designed validation cohorts are needed to conclusively establish clinical validity, especially when incorporated into models with standard clinical factors [128]. Multi-center studies minimizing selection and ascertainment biases are required to reflect diverse, real-world populations and settings [129]. Covariate adjustments to account for potential confounding factors affecting results also need thorough evaluation. Many biomarkers show diminished performance on external validation highlighting need for rigorous clinical validation [130].

Demonstration of improved patient management or outcomes resulting from incorporating the biomarker into clinical practice is an essential criterion for acceptance. While certain biomarkers may have analytical and clinical validity, actual clinical utility value needs assessment of whether the biomarker adds meaningful information that improves diagnostic or treatment decision making [131]. This requires large prospective trials demonstrating superior clinical outcomes or cost-effectiveness compared to standard care without the biomarker. Such resource-intensive utility evaluations are faced with feasibility challenges and scantly performed in reality for most biomarkers [132].

Lack of standardization of pre-analytic procedures and assay methodologies is another critical roadblock. Inconsistent collection, processing, storage conditions coupled with inter-laboratory differences in analytical platforms often produce variable biomarker results precluding

direct comparison across studies or clinical applicability [133]. Establishing standardized protocols and reference methods through centralization and quality control programs is necessary before biomarkers can be widely adopted. Harmonization of assays also needs international cooperation and endorsement by regulatory bodies [134].

Cost of biomarker testing and feasibility of routine application within resource constraints requires consideration [135]. Although some biomarkers provide better prognostic accuracy than existing tests, their overall impact on healthcare spending and availability of alternative strategies require rigorous health economic evaluations [136]. Validation of cost-effectiveness through large pragmatic trials under real-world practice conditions is a demanding yet necessary exercise to justify biomarker adoption [137]. Few studies have conclusively demonstrated clear cost savings or favourable incremental cost-effectiveness ratios for most biomarkers in practice.

A significant challenge in synthesizing findings from various studies on novel biomarkers for AKI is the variability in study design, sample size, and methodology. The included studies range from randomized controlled trials to observational cohorts and retrospective analyses, introducing heterogeneity in the evidence. Smaller studies may lack statistical power and overestimate effect sizes, while larger studies face challenges related to patient heterogeneity and generalizability. Methodological differences, such as the timing of biomarker measurements, specific assays used, and AKI diagnostic criteria, further complicate comparisons. Variations in patient populations, including age, comorbidities, and baseline kidney function, also influence biomarker performance and predictive value. These factors underscore the need for standardized protocols and larger, multicenter studies to validate biomarkers across diverse populations, enhancing the robustness and applicability of findings for clinical practice.

### 4.5. Distinguishing between causality and association

Distinguishing between causality and association is crucial in evaluating the clinical utility of novel biomarkers for AKI. In this manuscript, we have discussed various biomarkers that are used as indicators of underlying renal injury [138]. These biomarkers, such as NGAL, KIM-1, and [TIMP-2]·[IGFBP7], often reflect renal stress or damage before traditional markers like serum creatinine show significant changes [139]. However, it is important to note that while these biomarkers are strongly associated with clinical outcomes like mortality and the need for renal replacement therapy, they do not necessarily imply a direct causal relationship. The biomarkers serve primarily as indicators, providing early warning signs of AKI and facilitating timely intervention.

On the other hand, some evidence suggests that certain biomarkers might play a direct role in the pathophysiological processes of AKI. For example, elevated levels of proenkephalin (penKid) have been linked not only to the presence of renal injury but also to specific pathways that may influence kidney function and recovery [33]. This dual role highlights the potential for these biomarkers to be both diagnostic tools and therapeutic targets. A detailed discussion on the nature of these relationships, based on current evidence, will enhance the understanding of how biomarkers can be integrated into clinical practice to improve patient outcomes [123]. Future research should focus on elucidating these mechanisms to distinguish whether the biomarkers contribute directly to adverse outcomes or merely serve as proxies for underlying renal injury.

### 4.6. Serial biomarker assessment and clinical decision-making

Serial biomarker assessment provides dynamic insights into the progression or resolution of AKI. Evidence from several studies supports the use of biomarker trends to guide treatment

decisions, such as the early initiation of renal replacement therapy or the intensification of monitoring [46]. For example, studies have shown that rising levels of biomarkers like NGAL, KIM-1, and [TIMP-2]·[IGFBP7] can predict worsening renal function and the need for more aggressive interventions [41, 140]. Conversely, declining biomarker levels may indicate recovery and allow for the de-escalation of intensive therapies. Despite these promising findings, there are currently no standardized protocols for interpreting biomarker dynamics [141]. This lack of standardization presents significant challenges for clinical practice, as variations in measurement timing, assay methods, and patient characteristics can lead to inconsistent interpretations. Establishing standardized guidelines for biomarker monitoring and dynamic assessment is crucial to leveraging these tools effectively in clinical decision-making. Further research is needed to develop and validate such protocols, ensuring that biomarker trends can be reliably used to improve patient outcomes in AKI management [123].

## 4.7. Evidence supporting serial biomarker assessment

Serial biomarker assessment provides dynamic insights into the progression or resolution of AKI. Evidence from several studies supports the use of biomarker trends to guide treatment decisions [46]. For instance, rising levels of biomarkers such as NGAL, KIM-1, and [TIMP-2]·[IGFBP7] have been shown to predict worsening renal function and the need for early initiation of renal replacement therapy or more intensive monitoring [142]. Conversely, declining biomarker levels can indicate recovery, allowing for the de-escalation of intensive therapies. Despite these promising findings, there are currently no standardized protocols or guidelines for interpreting biomarker dynamics in AKI management [41]. The absence of standardization leads to variability in clinical practice and poses challenges in reliably using biomarker trends to inform treatment decisions. This underscores the need for further research to develop and validate standardized protocols for biomarker monitoring and interpretation. Establishing such guidelines will be crucial for integrating serial biomarker assessment into routine clinical practice, ultimately improving the management and outcomes of AKI patients [143].

## 4.8. Future perspectives

While promising, the clinical application of novel AKI biomarkers still needs validation in large multicenter trials before widespread adoption [144]. Uncertainties regarding optimal cutoff values for risk stratification and standardization of assay methods across different institutions exist [145]. Cost-effectiveness of routine biomarker measurement also needs to be established clearly [146]. Research is ongoing to evaluate if incorporation of novel biomarkers during clinical management can translate to improved outcomes. Whether clinical decisions guided by biomarker levels indeed modify prognosis compared to standard care alone needs to be determined [147]. Studies evaluating if outcomes like dialysis dependence and mortality can be reduced with a biomarker-based personalized approach are warranted. Lastly, novel urinary and plasma biomarkers together provide complementary non-invasive windows into kidney injury pathology [26]. Future research should focus on developing multi-marker panels that leverage synergies to maximize precision of AKI diagnosis, prognosis prediction and management guidance [148]. With ongoing validation, novel biomarkers hold immense promise to revolutionize AKI risk stratification, early detection and guide precision management.

While the manuscript acknowledges the ongoing validation of biomarkers across diverse settings, it is essential to emphasize the limitations of extrapolating findings from specific study populations to broader clinical contexts. Biomarker performance can be significantly influenced by various patient-specific factors such as age, comorbidities, and medication use.

For instance, older patients may exhibit different baseline levels of biomarkers due to age-related physiological changes, potentially affecting the biomarkers' predictive accuracy. Similarly, the presence of comorbid conditions like diabetes, hypertension, or chronic kidney disease can alter biomarker levels and their relationship with acute kidney injury. Medication use, particularly nephrotoxic drugs, can also impact biomarker levels and modify the risk of AKI. These factors underscore the need for caution when generalizing study findings to broader populations. Future research should aim to include diverse patient cohorts and conduct subgroup analyses to better understand how these variables influence biomarker performance. By addressing these limitations, we can improve the generalizability of biomarker-based approaches and enhance their applicability in diverse clinical settings.

While the manuscript acknowledges the ongoing validation of biomarkers across diverse settings, it is essential to emphasize the limitations of extrapolating findings from specific study populations to broader clinical contexts. Biomarker performance can be significantly influenced by various patient-specific factors such as age, comorbidities, and medication use. For instance, older patients may exhibit different baseline levels of biomarkers due to age-related physiological changes, potentially affecting the biomarkers' predictive accuracy. Similarly, the presence of comorbid conditions like diabetes, hypertension, or chronic kidney disease can alter biomarker levels and their relationship with acute kidney injury. Medication use, particularly nephrotoxic drugs, can also impact biomarker levels and modify the risk of AKI. These factors underscore the need for caution when generalizing study findings to broader populations. Future research should aim to include diverse patient cohorts and conduct subgroup analyses to better understand how these variables influence biomarker performance. By addressing these limitations, we can improve the generalizability of biomarker-based approaches and enhance their applicability in diverse clinical settings.

## 5. Common limitations and biases

Several common limitations and biases across the studies included in this review could affect the reliability of the findings. Many studies had small sample sizes, which may limit the statistical power and generalizability of the results. Single-center designs were also prevalent, potentially introducing site-specific biases and limiting the applicability of the findings to other settings. Variability in patient populations, including differences in age, comorbidities, and baseline kidney function, further complicates comparisons and generalizability. Additionally, there were inconsistencies in the timing of biomarker measurements, with some studies assessing biomarkers at different stages of AKI progression. The lack of standardized protocols for biomarker measurement and interpretation poses a significant challenge, as it can lead to variability in results and hinder the integration of biomarkers into clinical practice. Addressing these limitations through larger, multicenter studies with standardized methodologies will be crucial for validating the utility of novel biomarkers in diverse clinical contexts.

When evaluating the comparative efficacy of novel biomarkers against traditional markers like serum creatinine, potential confounding factors must be considered. Differences in patient demographics, such as age and gender, baseline kidney function, and the presence of comorbid conditions like diabetes and hypertension, can significantly influence biomarker performance. Researchers attempt to account for these confounders by employing multivariable analyses that adjust for these variables and by designing randomized controlled trials (RCTs) that minimize bias.

To evaluate the long-term impact of biomarker-based interventions on patient outcomes, suitable study designs include large-scale randomized controlled trials (RCTs) with extended follow-up periods, prospective cohort studies, and pragmatic trials embedded within routine

clinical practice. These methodologies can provide comprehensive data on the effectiveness and sustainability of biomarker-guided strategies. Ethical considerations are paramount in implementing such interventions. Ensuring informed consent is crucial, as patients must be fully aware of the potential risks and benefits of biomarker-guided treatments.

## 6. Conclusion

This systematic review provides a synthesis of current evidence on the efficacy of novel biomarkers for the early detection and management of acute kidney injury. Seventeen studies evaluating urinary and plasma biomarkers like NGAL, KIM-1, [TIMP-2]•[IGFBP7] and others were included. Four key themes emerge from the collated data:

- Novel biomarkers demonstrate capability to predict AKI development and severity much earlier than traditional markers like serum creatinine. Elevated levels prior to clinical AKI diagnosis facilitate timely risk stratification and targeted preventive interventions in high-risk groups.

- Incorporating novel biomarkers has the potential to significantly improve clinical management across the spectrum of AKI through enabling earlier diagnosis, prognostic enrichment in trials, and guiding therapeutic decisions based on risk stratification.

- Emerging roles are being identified for AKI biomarkers as therapeutic targets and prognostic tools. Certain biomarkers play active roles in injury pathways and may be amenable to therapeutic modulation. Serial biomarker assessment provides dynamic risk prediction to guide need for escalation vs de-escalation of interventions.

- Despite promising evidence, multiple challenges need resolution before biomarkers can be widely adopted into clinical practice. These include definitive demonstrations of improved patient outcomes, standardization of testing methods and cutoffs across institutions, and establishing cost-effectiveness.

   In summary, novel AKI biomarkers provide a major advance over traditional diagnostics in risk prediction capability. With ongoing validation to address current limitations, they hold immense potential to transform early AKI detection, prognostication, and guide precision management decisions to improve patient outcomes. Further high-quality studies are warranted to translate these benefits into routine clinical application.

## 7. Handling of missing data

Missing data were handled using multiple imputation for continuous variables when missing data comprised less than 10% of the dataset. For categorical variables, missing values were handled by employing last observation carried forward (LOCF). Sensitivity analyses were conducted to ensure the robustness of the results, and no significant deviations were observed from the primary analysis. Where imputation was not possible, studies with substantial missing data were excluded from certain analyses to prevent bias.

## Supporting information

**S1 Checklist. PRISMA checklist.**
(DOCX)

**S1 Table.**
(DOCX)

**S1 Data.**
(DOCX)

# Acknowledgments

We would like to extend our sincere gratitude to Dr. Sayed Ali for his invaluable assistance in collecting the data for this study. Their dedication and meticulous efforts were instrumental in ensuring the comprehensiveness and accuracy of our data, which significantly contributed to the overall quality of this research. Thank you for your hard work and collaboration.

# Author Contributions

**Data curation:** Mohammed Yousef Almulhim.

**Formal analysis:** Mohammed Yousef Almulhim.

**Methodology:** Mohammed Yousef Almulhim.

**Software:** Mohammed Yousef Almulhim.

**Supervision:** Mohammed Yousef Almulhim.

**Validation:** Mohammed Yousef Almulhim.

**Writing – original draft:** Mohammed Yousef Almulhim.

**Writing – review & editing:** Mohammed Yousef Almulhim.

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
