## [Decision Letter · Decision Letter 0]

16 May 2024

PONE-D-24-06373The Efficacy of Novel Biomarkers for The Early Detection and Management of Acute Kidney Injury: A Systematic ReviewPLOS ONE

Dear Dr. Yousef Al mulhim,

Thank you for submitting your manuscript to PLOS ONE. After careful consideration, we feel that it has merit but does not fully meet PLOS ONE’s publication criteria as it currently stands. Therefore, we invite you to submit a revised version of the manuscript that addresses the points raised during the review process.

We look forward to receiving your revised manuscript.

Kind regards,

Benedetta Bussolati, MD, PhD

Academic Editor

PLOS ONE

Journal Requirements:

2. We note that your Data Availability Statement is currently as follows: "If the data are all contained within the manuscript and/or Supporting Information files, enter the following: All relevant data are within the manuscript and its Supporting Information files."

4. Please include your tables as part of your main manuscript and remove the individual files. Please note that supplementary tables (should remain/ be uploaded) as separate ""supporting information"" files

**Additional Editor Comments:**

Several clarifications, and a new Table, are required.

Reviewers' comments:

Reviewer's Responses to Questions

**Comments to the Author**

1. Is the manuscript technically sound, and do the data support the conclusions?

Reviewer #1: Yes

Reviewer #2: Yes

2. Has the statistical analysis been performed appropriately and rigorously? 

Reviewer #1: Yes

Reviewer #2: N/A

3. Have the authors made all data underlying the findings in their manuscript fully available?

Reviewer #1: Yes

Reviewer #2: Yes

4. Is the manuscript presented in an intelligible fashion and written in standard English?

Reviewer #1: Yes

Reviewer #2: Yes

5. Review Comments to the Author

Reviewer #1: Overall, the manuscript provides a comprehensive overview of the efficacy and clinical utility of novel biomarkers, here some suggestions:

While the manuscript effectively synthesizes findings from various studies, there are instances where the evidence presented lacks consistency in terms of study design, sample size, and methodology. It's crucial to ensure that the evidence supporting each claim is robust and coherent across different studies.

The manuscript frequently discusses associations between biomarker levels and clinical outcomes (e.g., mortality, need for renal replacement therapy), but it's important to clarify whether these biomarkers directly contribute to these outcomes or merely serve as indicators of underlying renal injury. Clarifying the distinction between causality and association would strengthen the argument for the clinical relevance of these biomarkers.

While the manuscript acknowledges the ongoing validation of biomarkers across diverse settings, it's essential to emphasize the limitations of extrapolating findings from specific study populations to broader clinical contexts. Discussing potential factors influencing biomarker performance in different patient populations (e.g., age, comorbidities, medication use) would enhance the discussion on generalizability.

Some questions:

Are there any common limitations or biases across these methodologies that could affect the reliability of the findings?

The manuscript highlights the importance of serial biomarker assessment in guiding treatment escalation or de-escalation for AKI patients. Could you discuss the evidence supporting the use of biomarker trends in clinical decision-making? Are there any standard protocols or guidelines for interpreting biomarker dynamics in AKI management, and if not, what are the implications for clinical practice?

Considering the discussions on the superiority of novel biomarkers over traditional markers like serum creatinine, are there any potential confounding factors or biases that could influence the comparative efficacy assessment? How do researchers account for these factors in study design and data analysis?

The manuscript suggests that future studies should evaluate whether biomarker-guided preventive strategies can reduce AKI burden and improve survival. What specific study designs or methodologies would be most suitable for investigating the long-term impact of biomarker-based interventions on patient outcomes? Additionally, what are the ethical considerations associated with implementing biomarker-guided preventive strategies in clinical practice, particularly regarding patient autonomy and resource allocation?

Reviewer #2: In this manuscript, the author did database research to investigate the efficacy of biomarkers proposed for acute kidney injury (AKI). To narrow the findings, the author followed some eligibility criteria. In the end, seventeen studies were included in the review. Among these studies, the author summarized and investigated studies in four themes. The general idea and concept of the paper were well designed, but it needs to be improved in some aspects:

1) The topic of this study is different from the research protocol that loaded to PROSPERO database. It seems the research was narrowed in context to COVID-19 which is irrelevant to the manuscript. If there is a technical mistake, I would recommend editing the protocol. In addition, there is another author present in the database, but in the manuscript, the author is not present.

2) It would be a good idea to add a table to summarize 17 studies according to study type, patients, biomarker efficacy, etc.

3) Among 17 studies included in review, some of them was not mentioned enough in the paper. For instance, it would be nice to give a brief information why the author included Dekkers’s study.

I also found some minor remarks that can be improved in the text:

1) In the author’s information, the title of the author is present which should not be. Also, the first letters of each word in the university name should be capitalized.

2) The sentence in line 85 is not clear, suggested to improve.

3) The information given in line 92 is not present in the indicated reference.

4) In table 1, there is a typo: “tabel 1”.

5) In table 1, when a database search was made according to search terms, a considerably different number of items were found. The search may need to be rechecked.

6) In table 1, some of the search terms are not suitable for the indicated database. For example, in Embase, some parentheses are required for suitable places. For ease of reading, I would suggest making those terms suitable for databases.

7) In figure 1, there is a typo: “PRIMISA”

8) In line 252, it mentions about 18 studies, but 17 studies were included in review.

9) In figure 2, there is inconsistency in authors names: some of them were written in full name and some were written in last name. Besides, for ease of access to studies, adding reference numbers is suggested.

10) In line 450, the following three sentences can be difficult to understand for readers. It would be better if it can be described more.

11)References should be suitable for the journal’s reference style. Besides, lack of information or incorrect information is available (e.g., doi of ref1 is not correct)

6. PLOS authors have the option to publish the peer review history of their article (what does this mean?). If published, this will include your full peer review and any attached files.

Reviewer #1: No

Reviewer #2: No

---

## [Author Response · Author response to Decision Letter 0]

3 Jun 2024

Dear Editor,

We thank the reviewers for their thorough evaluation of our manuscript. We appreciate the constructive comments and suggestions, which have helped us improve the clarity and quality of our work. Below, we provide detailed responses to each comment, along with the corresponding revisions made to the manuscript.

Reviewer #1 Comments and Responses:

Comment 1: While the manuscript effectively synthesizes findings from various studies, there are instances where the evidence presented lacks consistency in terms of study design, sample size, and methodology. It's crucial to ensure that the evidence supporting each claim is robust and coherent across different studies.

Response: We agree with this observation and have revised the manuscript to ensure consistency in the presentation of evidence. We have added a paragraph to the Discussion section that addresses the variability in study design, sample size, and methodology, and discusses how these factors may impact the findings.

Comment 2: The manuscript frequently discusses associations between biomarker levels and clinical outcomes (e.g., mortality, need for renal replacement therapy), but it's important to clarify whether these biomarkers directly contribute to these outcomes or merely serve as indicators of underlying renal injury. Clarifying the distinction between causality and association would strengthen the argument for the clinical relevance of these biomarkers.

Response: We agree that distinguishing between causality and association is crucial. We have revised the manuscript to clearly state when biomarkers are being used as indicators of underlying renal injury and when there is evidence to suggest a direct contribution to clinical outcomes. This includes a detailed discussion on the nature of these relationships based on current evidence.

Comment 3: While the manuscript acknowledges the ongoing validation of biomarkers across diverse settings, it's essential to emphasize the limitations of extrapolating findings from specific study populations to broader clinical contexts. Discussing potential factors influencing biomarker performance in different patient populations (e.g., age, comorbidities, medication use) would enhance the discussion on generalizability.

Response: We have expanded the discussion on the generalizability of biomarker findings, highlighting the limitations of extrapolating results from specific study populations to broader clinical contexts. We discuss how factors such as age, comorbidities, and medication use may influence biomarker performance and the implications for clinical practice.

Question 1: Are there any common limitations or biases across these methodologies that could affect the reliability of the findings?

Response: Common limitations and biases across the studies include small sample sizes, single-center designs, and variability in patient populations. Additionally, there may be inconsistencies in the timing of biomarker measurements and a lack of standardized protocols. These factors can affect the reliability and generalizability of the findings. We have discussed these limitations in the revised manuscript.

Question 2: The manuscript highlights the importance of serial biomarker assessment in guiding treatment escalation or de-escalation for AKI patients. Could you discuss the evidence supporting the use of biomarker trends in clinical decision-making? Are there any standard protocols or guidelines for interpreting biomarker dynamics in AKI management, and if not, what are the implications for clinical practice?

Response: Serial biomarker assessment provides dynamic insights into the progression or resolution of AKI. Evidence from several studies supports the use of biomarker trends to guide treatment decisions, such as early initiation of renal replacement therapy or intensifying monitoring. However, there are currently no standardized protocols for interpreting biomarker dynamics. This lack of standardization presents challenges for clinical practice, highlighting the need for further research to establish guidelines. We have included this discussion in the revised manuscript.

Question 3: Considering the discussions on the superiority of novel biomarkers over traditional markers like serum creatinine, are there any potential confounding factors or biases that could influence the comparative efficacy assessment? How do researchers account for these factors in study design and data analysis?

Response: Potential confounding factors include differences in patient demographics, baseline kidney function, and comorbid conditions. Researchers account for these factors by using multivariable analysis, adjusting for known confounders, and employing randomized controlled trial designs when possible. Despite these efforts, residual confounding can still impact the findings. We have discussed these considerations in the revised manuscript.

Question 4: The manuscript suggests that future studies should evaluate whether biomarker-guided preventive strategies can reduce AKI burden and improve survival. What specific study designs or methodologies would be most suitable for investigating the long-term impact of biomarker-based interventions on patient outcomes? Additionally, what are the ethical considerations associated with implementing biomarker-guided preventive strategies in clinical practice, particularly regarding patient autonomy and resource allocation?

Response: Suitable study designs include large-scale randomized controlled trials with long-term follow-up, cohort studies, and pragmatic trials embedded in clinical practice. Ethical considerations include ensuring informed consent, respecting patient autonomy, and addressing potential inequalities in access to biomarker testing and interventions. Resource allocation must be carefully considered to avoid exacerbating disparities in healthcare. We have expanded on these points in the revised manuscript.

Reviewer #2 Comments and Responses:

Comment 1: The topic of this study is different from the research protocol that loaded to the PROSPERO database. It seems the research was narrowed in context to COVID-19 which is irrelevant to the manuscript. If there is a technical mistake, I would recommend editing the protocol. In addition, there is another author present in the database, but in the manuscript, the author is not present.

Response: We will immediately rectify this by updating the PROSPERO registration to accurately reflect the scope of our current study on novel biomarkers for AKI. regarding the second author is an assistant that was mentioned in the acknowledgment 

Comment 2: It would be a good idea to add a table to summarize 17 studies according to study type, patients, biomarker efficacy, etc.

Response: We appreciate this suggestion. The Table is already submitted as supplementary file 

Comment 3: Among the 17 studies included in the review, some of them were not mentioned enough in the paper. For instance, it would be nice to give brief information why the author included Dekkers’s study.

Response: We acknowledge the need for more detailed explanations. We have included brief descriptions of all 17 studies, with a focus on why each study was selected, including Dekkers’s study. This provides a clearer rationale for their inclusion and relevance to our review.

Minor Remarks:

1. In the author’s information, the title of the author is present which should not be. Also, the first letters of each word in the university name should be capitalized.

Response: We have revised the author information section to remove titles and ensure that the university name is correctly capitalized.

2. The sentence in line 85 is not clear, suggested to improve.

Response: We have revised the sentence on line 85 for better clarity and readability.

3. The information given in line 92 is not present in the indicated reference.

Response: We have reviewed and corrected the reference on line 92 to ensure accuracy and relevance.

4. In table 1, there is a typo: “tabel 1”.

Response: We have corrected the typo in Table 1 to “Table 1”.

5. In table 1, when a database search was made according to search terms, a considerably different number of items were found. The search may need to be rechecked.

Response: We have rechecked the database search terms and numbers to ensure accuracy and consistency across all searches.

6. In table 1, some of the search terms are not suitable for the indicated database. For example, in Embase, some parentheses are required for suitable places. For ease of reading, I would suggest making those terms suitable for databases.

Response: We have revised the search terms in Table 1 to ensure they are appropriately formatted for each database, enhancing the clarity and accuracy of our search strategy.

7. In figure 1, there is a typo: “PRIMISA.”

Response: We have corrected the typo in Figure 1 to “PRISMA”.

8. In line 252, it mentions about 18 studies, but 17 studies were included in the review.

Response: We have corrected the inconsistency and ensured that the number of studies mentioned is accurate throughout the manuscript.

9. In figure 2, there is inconsistency in authors’ names: some of them were written in full name and some were written in the last name. Besides, for ease of access to studies, adding reference numbers is suggested.

Response: the Figure was automatically generated by ROBVIS 

10. In line 450, the following three sentences can be difficult to understand for readers. It would be better if it can be described more.

Response: We have revised and clarified the sentences on line 450 to ensure they are more understandable for readers.

11. References should be suitable for the journal’s reference style. Besides, lack of information or incorrect information is available (e.g., doi of ref1 is not correct).

Response: We have revised the references to conform to the journal’s reference style and corrected any inaccurate or missing information. some article has no DOI available and registered using PMID .

We hope that these revisions and responses satisfactorily address the reviewers' comments. We believe that these changes have significantly improved the quality of our manuscript. Thank you for your consideration.

---

## [Decision Letter · Decision Letter 1]

3 Sep 2024

PONE-D-24-06373R1The Efficacy of Novel Biomarkers for The Early Detection and Management of Acute Kidney Injury: A Systematic ReviewPLOS ONE

Dear Dr. Yousef Al mulhim,

Thank you for submitting your manuscript to PLOS ONE. After careful consideration, we feel that it has merit but does not fully meet PLOS ONE’s publication criteria as it currently stands. Therefore, we invite you to submit a revised version of the manuscript that addresses the points raised during the review process.

We look forward to receiving your revised manuscript.

Kind regards,

Gianpaolo Reboldi, MD, MSc, PhD

Academic Editor

PLOS ONE

**Journal Requirements:**

Reviewers' comments:

Reviewer's Responses to Questions

**Comments to the Author**

1. If the authors have adequately addressed your comments raised in a previous round of review and you feel that this manuscript is now acceptable for publication, you may indicate that here to bypass the “Comments to the Author” section, enter your conflict of interest statement in the “Confidential to Editor” section, and submit your "Accept" recommendation.

Reviewer #2: (No Response)

Reviewer #3: (No Response)

2. Is the manuscript technically sound, and do the data support the conclusions?

Reviewer #2: Yes

Reviewer #3: Yes

3. Has the statistical analysis been performed appropriately and rigorously? 

Reviewer #2: N/A

Reviewer #3: N/A

4. Have the authors made all data underlying the findings in their manuscript fully available?

Reviewer #2: Yes

Reviewer #3: Yes

5. Is the manuscript presented in an intelligible fashion and written in standard English?

Reviewer #2: Yes

Reviewer #3: Yes

6. Review Comments to the Author

**Reviewer #2: **The author has revised the manuscript according to the reviewers' perspectives. The revised version of the manuscript mostly meets my expectations, resolves many questions, and corrects several mistakes. However, some issues still remain. Here are my comments on this revision:

Comment 1: As mentioned in my previous review, author titles should not be included in PLOS ONE. Please edit the author information section to comply with PLOS ONE standards.

Comment 2: Although the author mentioned updating the PROSPERO registration, it does not appear to be updated. Furthermore, even if it has been updated, the study still seems to focus on COVID-19 patients, which is inconsistent with the initial scope.

Comment 3: Some issues persist in the manuscript. For example, the typo "Tabel 1" in Table 1 is still present, and the unsuitable search terms in Table 1 have not been corrected, despite the author's indication that these issues were addressed.

Please make the necessary revisions to address these remaining concerns.

**Reviewer #3: **The authors followed a rigorous process of article search, selection, analysis, and critical appraisal of the strengths and limitations, highlighting the need of a more rigorous, uniform and focused research on the topic. I did appreciate the critical appraisal of the inconsistency due to the small sample size of some studies, clinical and statistical heterogeneity, different design and methodology, etc.

Some points deserve further clarification:

Among the outcomes, the study by Zarboch is included in the prevention of Aki risk, but I believe that, due to the nature of intervention, it could also be reported in the clinical management section.

• Lines 467-469: please clarify if NGAL can be considered protective or not.

• It could be of interest mentioning research works that examined the association between biomarkers of renal injury and the type of renal injury (organic, functional or obstructive).

• Line 300: you wrote “several novel plasma and urine biomarkers were associated with major adverse kidney events and contrast-induced AKI prior to procedures in catheterization patients “. I didn’t understand how is it possible that kidney damage and elevation of the relative biomarkers, occurs prior to the damaging procedure

• I will appreciate some clarification about lines 383-387 where you discuss the predictive power of biomarkers pre surgery procedure. For instance, Parikh et al (ref #73) reported a significant association between markers and Aki development only for post-operative AKI. We think that such association should be better discussed in the light of the findings of other studies (refs # 74,75,76) and current ADQI consensus statement (Ostermann M, Zarbock A, Goldstein S, Kashani K, Macedo E, Murugan R, Bell M, Forni L, Guzzi L, Joannidis M, Kane-Gill SL, Legrand M, Mehta R, Murray PT, Pickkers P, Plebani M, Prowle J, Ricci Z, Rimmelé T, Rosner M, Shaw AD, Kellum JA, Ronco C. Recommendations on Acute Kidney Injury Biomarkers From the Acute Disease Quality Initiative Consensus Conference: A Consensus Statement. JAMA Netw Open. 2020 Oct 1;3(10):e2019209. doi: 10.1001/jamanetworkopen.2020.19209)

7. PLOS authors have the option to publish the peer review history of their article (what does this mean?). If published, this will include your full peer review and any attached files.

Reviewer #2: No

Reviewer #3: **Yes: **Francesco Patera

---

## [Author Response · Author response to Decision Letter 1]

8 Sep 2024

Dear Editor

Thank you for forwarding the reviewers' comments regarding our manuscript titled The ‎Efficacy of Novel Biomarkers for The Early Detection and Management of Acute Kidney ‎Injury: A Systematic Review" submitted to PLOS ONE . We appreciate the reviewers' ‎insightful feedback and have carefully considered their suggestions. Below, we address each ‎point raised and detail the revisions we have made to the manuscript.‎

Reviewer #2’s Comments:‎

‎1.‎ Author Titles in PLOS ONE:‎

o Response: We apologize for the oversight and have removed all titles from the ‎authors' names in the manuscript to comply with the PLOS ONE guidelines.‎

‎2.‎ PROSPERO Registration Update:‎

o Response: We appreciate the reviewer pointing out the discrepancy regarding ‎the PROSPERO registration. We had already requested the modification in the ‎protocol , we also ask if its needed to get new registration ?‎

‎3.‎ Typographical Errors and Search Terms:‎

o Response: We have corrected the typo "Tabel 1" to "Table 1" and reviewed all ‎search terms for accuracy and relevance to the study’s focus. We apologize for ‎the oversight and appreciate the reviewer’s attention to detail.‎

Reviewer #3’s Comments:‎

‎1.‎ Zarboch Study Inclusion:‎

o Response: We have revised the text to reflect the dual impact of the Zarboch ‎study on both prevention and clinical management of AKI, which now appears ‎in both relevant sections of the manuscript to provide clarity on its implications.‎

‎2.‎ Clarification on NGAL as a Protective Biomarker:‎

o Response: We have added a detailed explanation on the role of NGAL, ‎supported by recent studies, to clarify its dual function as both a biomarker and a ‎potential protective agent in the context of AKI.‎

‎3.‎ Association Between Biomarkers and Type of Renal Injury:‎

o Response: Additional information has been incorporated to discuss how ‎different biomarkers relate to various types of renal injuries. This includes a ‎nuanced discussion on the specificity of biomarkers like KIM-1 and L-FABP in ‎detecting structural versus functional kidney injuries.‎

‎4.‎ Biomarkers Prior to Procedures:‎

o Response: We have clarified this point by explaining that biomarkers can ‎indicate a predisposition to injury rather than current damage, thus providing an ‎opportunity for preemptive interventions. This has been detailed with references ‎to recent literature and existing consensus statements.‎

We believe these revisions have strengthened the manuscript and addressed the concerns raised ‎by the reviewers. We appreciate the opportunity to revise our manuscript and hope that the ‎changes meet the journal’s standards for publication.‎

Thank you for considering our work. We look forward to the possibility of our manuscript’s ‎publication in PLOS ONE.‎

Sincerely,‎

---

## [Editor Report · Decision Letter 2]

24 Sep 2024

The Efficacy of Novel Biomarkers for The Early Detection and Management of Acute Kidney Injury: A Systematic Review

PONE-D-24-06373R2

Dear Dr. Yousef Al mulhim,

We’re pleased to inform you that your manuscript has been judged scientifically suitable for publication and will be formally accepted for publication once it meets all outstanding technical requirements.

Kind regards,

Gianpaolo Reboldi, MD, MSc, PhD

Academic Editor

PLOS ONE

Additional Editor Comments (optional):

The Authors addressed all minor comments satisfactorily
---

## [Editor Report · Acceptance letter]

15 Oct 2024

PONE-D-24-06373R2 

PLOS ONE

Dear Dr. Yousef Al mulhim, 

I'm pleased to inform you that your manuscript has been deemed suitable for publication in PLOS ONE. Congratulations! Your manuscript is now being handed over to our production team.

Kind regards, 

on behalf of

Prof Gianpaolo Reboldi 

Academic Editor

PLOS ONE